# Impact of Preference Noise on the Alignment Performance of Generative Language Models

**Yang Gao, Dana Alon & Donald Metzler**
Google DeepMind
{gaostayyang,danama,metzler}@google.com

## Abstract

A key requirement in developing Generative Language Models (GLMs) is to have their values aligned with human values. *Preference-based alignment* is a widely used paradigm for this purpose, in which *preferences* over generation pairs are first elicited from human annotators or AI systems, and then fed into some alignment techniques, e.g., Direct Preference Optimization. However, a substantial percent (20 - 40%) of the preference pairs used in GLM alignment are *noisy*, and it remains unclear how the noise affect the alignment performance and how to mitigate their negative impact. In this paper, we propose a framework to inject desirable amounts and types of noise to the preferences, and systematically study the impact of preference noise on the alignment performance in two tasks (summarization and dialogue generation). We find that the alignment performance can be highly sensitive to the noise rates in the preference data: e.g., a 10 percentage points (pp) increase of the noise rate can lead to 30 pp drop in the alignment performance (in win rate). To mitigate the impact of noise, *confidence-based data filtering* shows significant benefit when certain types of noise are present. We hope our work can help the community better understand and mitigate the impact of preference noise in GLM alignment.

## 1 Introduction

As the capabilities of *Generative Language Models* (GLMs) keep improving through pre-training at a large scale, methods for *aligning GLMs with human preferences*, i.e., steering GLMs to follow user instructions effectively and safely, have attracted increasing attention (Ji et al., 2023). A widely used paradigm to align GLMs with human values is to first collect *binary preferences on generation pairs*, and then use techniques like *Proximal Policy Optimization* (a Reinforcement Learning algorithm, (Schulman et al., 2017)), *Direct Preference Optimization* (DPO, (Rafailov et al., 2023)), or *Sequence Likelihood Calibration* (SLiC, (Zhao et al., 2023)) to align the GLMs with the collected preferences. The binary preferences can be provided by human annotators, trained *Reward Models* (RMs, (Ouyang et al., 2022)), or *Constitutional AI agents* (Bai et al., 2022b; Lee et al., 2024). Preference-based GLM alignment has proven to be highly effective in improving the safety and usability of GLMs, and hence has been used to develop both open-source (Touvron et al., 2023; Gemma, 2024) and proprietary (Google, 2023; Achiam et al., 2023) GLMs.

However, the binary preferences used in GLM alignment are often *noisy*, i.e., containing preferences that disagree with the ground truth (e.g., preferences provided by domain experts). Zheng et al. (2023) report that 19-37% preferences provided by crowd workers are noisy. Similar noise rates are also observed in preferences provided by RMs and Constitutional AI. Table 1 summarizes the noise rates of preferences used in recent GLM alignment works. It is generally believed that the lower the noise rates in the preferences, the better the final alignment performance (Lee et al., 2024), but it remains unclear what is the *quantitative relation between noise rates and alignment performance*, and, furthermore, how to *mitigate the negative impact of preference noise* on alignment performance. In this paper, we answer these questions with systematic empirical studies.

| Oracle | Task | Noise% | Reference |
|---|---|---|---|
| Human | MTBench | 19-37 | (Zheng et al., 2023) |
| Constitutional AI | MTBench | 15-34 | (Zheng et al., 2023) |
| | TL;DR | 22 | (Lee et al., 2024) |
| | CBArena | 22-36 | (Zheng et al., 2023) |
| | AntHH | 27.9-30.9 | (Lee et al., 2024) |
| | MetaHS | 41.4-41.9 | (Touvron et al., 2023) |
| Reward Models | TL;DR | 21.3-27 | (Zhao et al., 2023; Munos et al., 2023) |
| | SHP | 26.3 | (Cui et al., 2023) |
| | MetaHS | 35.5-36.8 | (Touvron et al., 2023) |
| | WebGPT | 34.8 | (Cui et al., 2023) |

Table 1: Preference noise are observed in a wide range of tasks, including video game (Atari), QA (MTBench, **S**tanford**H**uman**P**reference), Summarization (TL;DR), and Dialogue (WebGPT, **C**hat**B**ot**A**rena, **Ant**ropic**H**elpful**H**armless, **Meta**H**elpful**S**afety).

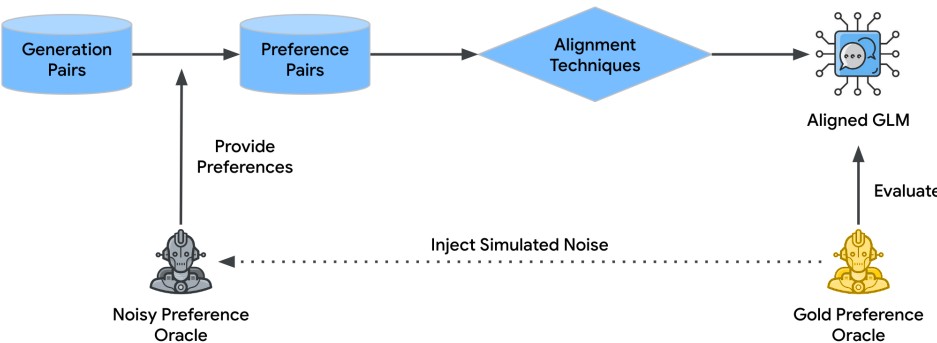

Figure 1: Our framework for evaluating the impact of preference noise on GLM alignment.

There exist frameworks for studying the influence of preference noise on alignment performance in Robotics, e.g., the *B-Pref* framework by Lee et al. (2021a). They assume a gold reward model is available, and design different strategies to *corrupt* the gold reward model to provide (simulated) noisy preferences (see §2 for more details). Although they have been successfully used to benchmark and compare different preference-based RL algorithms in Robotics tasks, they are not applicable for GLM alignment, mainly for two reasons: **(i)** some of their noise simulation strategies are unsuitable for NLP tasks; and **(ii)** the alignment techniques used in Robotics (e.g., PREBBLE (Lee et al., 2021b)) are different from those used in GLM (e.g., DPO and SLiC). To alleviate these problems, we propose a new framework whose noise-simulation strategies and alignment techniques are tailored for GLM alignment. Figure 1 illustrates the framework.

With the proposed framework, we perform controlled experiments to study the impact of preference noise on alignment performance on two tasks, summarization (Stiennon et al., 2020) and dialogue generation (Bai et al., 2022a). We find that even with high (45%) noise rates, GLM alignment is still beneficial (i.e., yielding 50%+ win rate). However, alignment performance is also highly sensitive to noise rates: a 10 percentage points (pp) increase of noise rates can lead to 30 pp drop in the alignment performance (in terms of win rate). We also explore different strategies to mitigate the negative impact of preference noise on alignment performance. We find that some widely used *regularization* methods fail to mitigate the negative impact, but *confidence-based data selection* can effectively improve performance in realistic settings. We hope our findings can help GLM developers better understand the impact of preference noise on alignment performance, and that our framework can facilitate the exploration of more effective and noise-robust alignment methods.

## 2 Related Works

**Preference-based GLM alignment.** Pairwise preferences are widely used in AI alignment, because controlled experiments have suggested that asking for preferences places a lower cognitive burden on the human subjects than asking for absolute ratings or categorized labels (Kendall, 1948; Thurstone, 2017; Kingsley & Brown, 2010). In GLM alignment, a common practice is to first train an RM from the human provided preferences, and then use the RM to provide reward signals in Reinforcement Learning (RL) (Böhm et al., 2019; Gao et al., 2020; Stiennon et al., 2020). Recent methods like DPO (Rafailov et al., 2023), SLiC (Zhao et al., 2023), and Identify Preference Optimization (IPO, Azar et al. (2023)) go one step further, by eliminating the RM training step and directly using preference pairs to train the final GLM. Azar et al. (2023) has shown that these methods essentially optimize the same learning objective, and they differentiate in their regularization terms and function approximation methods. Compared to the RM-RL two-stage paradigm, DPO and SLiC yield stronger performance in multiple NLP applications with lower computational costs (Chen et al., 2024; Yuan et al., 2024).

**Learning from noisy data.** Data used in real-world machine learning applications are often noisy (c.f., Song et al. (2022)), and deep neural models are particularly sensitive to data noise, as they are prone to overfit to noise patterns in the training data (Han et al., 2020). Hence, multiple methods have been proposed to improve the robustness of the neural models, mostly falling into four categories (Frénay & Verleysen, 2014; Song et al., 2022): *robust neural architectures*, *regularization methods*, *robust loss functions*, and *data filtering methods*. In this paper, we stick to the well established architecture (Transformers by Vaswani et al. (2017)) and loss functions (DPO), and explore different regularization and data selection methods to mitigate the negative impact of noisy preferences on the alignment performance.

The impact of noisy data on RL has also been studied. Lee et al. (2021a) propose the *B-Pref* benchmark, a platform to test the performance and robustness of preference-based RL algorithms in the face of different types of preference noise on various locomotion and robotic manipulation tasks. To simulate realistic preference noise, they assume they have access to a gold-standard reward model $r^*$, and design five strategies to derive noisy preferences therefrom, including the *stochastic strategy* ($p(y_0 > y_1) = \sigma[r^*(y_0) - r^*(y_1)]$, where $\sigma$ is the sigmoid function such that $\sigma(x) = (1 + \exp(-x))^{-1}$), *myopic strategy* (providing preferences only based on the last part of the presented candidates), *skipping strategy* (reject to provide preferences if both candidates are low-quality), *equally-preferable strategy* (when the quality of the presented candidates are similar, mark them as a tie), and *random mistake strategy* (randomly flip the correct preference direction with a fixed chance). We note that some of these strategies are unrealistic in GLM alignment, e.g., the myopic strategy (users usually do not judge the quality of texts based on their last parts), skipping strategy, and equally-preferable strategy (ties are usually not allowed in annotating text qualities). Also, they only consider noise from human annotators but ignore those from AI-based annotators (e.g., RLAIF (Lee et al., 2024)). For these reasons, we propose a new set of strategies for simulating preference noise in GLM alignment in §4.

## 3 Preliminaries

Let $\mathcal{X}$ be the set of all prompts, and $\mathcal{Y}_{\mathcal{X}}$ be the set of all possible continuations for all prompts in $\mathcal{X}$. We assume there exists a gold reward model $r^* : \mathcal{X} \times \mathcal{Y}_{\mathcal{X}} \to \mathbb{R}$, which measures the quality of continuation $y \in \mathcal{Y}_{\mathcal{X}}$ for prompt $x \in \mathcal{X}$ on some desired aspects (e.g., helpfulness, informativeness, or harmlessness). A GLM can be defined as a policy $\pi$, such that $\pi(y|x)$ is the probability of generating $y$ for the input prompt $x$. The objective of *GLM alignment* is to find the optimal policy that can maximize the expected gold reward value while minimizing the divergence from a reference policy:

$$\max_{\pi} \sum_{x \in \mathcal{X}, y \sim \pi(\cdot|x)} [r^*(y|x)] - \beta \mathbb{D}_{KL}[\pi_\theta(y|x)||\pi_{\text{sft}}(y|x)], \tag{1}$$

where $\beta$ is a hyperparameter, $\mathbb{D}_{KL}$ is the Kullback–Leibler divergence, and $\pi_{\text{sft}}$ is the reference policy.

In practice, we cannot optimize Eq. (1) directly, because the gold reward $r^*$ is usually inaccessible, and the summation operation is prohibitively expensive. Multiple algorithms have been proposed to obtain (approximate) solutions for the objective (see §2). In this work, we use DPO (Rafailov et al., 2023) because of its strong performance and lower computational cost compared to other methods (e.g., PPO). The loss function in DPO is:

$$\mathcal{L}(\pi_\theta) = -\mathbb{E}_{(x,y_w,y_l)\sim\mathcal{D}}\{\log\sigma[\beta\log(\frac{\pi_\theta(y_w|x)}{\pi_{\text{sft}}(y_w|x)}) - \beta\log(\frac{\pi_\theta(y_l|x)}{\pi_{\text{sft}}(y_l|x)})]\}, \qquad (2)$$

where $\pi_\theta$ is the learnable policy parameterized by $\theta$, $\sigma$ is the sigmoid function, and $\mathcal{D} = \{(x^i, y_w^i, y_l^i)\}_{i=1}^n$ is the training dataset consisting of $n$ data entries. Each data entry in $\mathcal{D}$ consists of a prompt $x^i \in \mathcal{X}$ and two continuations $y_w^i, y_l^i \in \mathcal{Y}_\mathcal{X}$, such that $y_w^i$ is preferred over $y_l^i$.

It has been proven that if $\mathcal{D}$ is sufficiently large and all pairs in $\mathcal{D}$ are noise-free (i.e., $r^*(y_w^i) > r^*(y_l^i)$ for $i = 1, \cdots, n$), the policy learned by DPO is (near-)optimal with respect to Eq. (1) (Rafailov et al., 2023; Azar et al., 2023). However, in practice, some preferences in $\mathcal{D}$ can be *noisy*, i.e., different from the preference direction induced by the gold reward model. In this work, we remove the (strong) noise-free assumption on $\mathcal{D}$, but instead introduce different rates and types of noise to $\mathcal{D}$ (in §4) and empirically study their impact on the quality of $\pi_\theta$ (in §6).

To measure the alignment performance (i.e., measure the performance of $\pi_\theta$), we compute the *win rate* between $\pi_\theta$ and $\pi_{\text{sft}}$:

$$w = \frac{1}{|\mathcal{X}_{\text{test}}|}\sum_{x\in\mathcal{X}_{\text{test}}}\mathbb{1}[r^*(x,y_{\pi_\theta}) > r^*(x,y_{\pi_{\text{sft}}})], \qquad (3)$$

where $\mathcal{X}_{\text{test}} \subset \mathcal{X}$ is a held-out test prompt set, and $y_{\pi_\theta}$ and $y_{\pi_{\text{sft}}}$ are generations sampled from $\pi_\theta$ and $\pi_{\text{sft}}$, respectively.

## 4 Noisy Preferences

The preferences are often *noisy*, i.e., disagree with the preference directions induced by the gold reward model $r^*$. Inspired by past works (Lee et al., 2021a) (see §2 for more discussions), we consider three *oracles* to provide different types of noisy preferences.

- **Random Noise Oracle**. When presented with a prompt $x \in \mathcal{X}$ and a pair of responses $y_w, y_l \in \mathcal{Y}_\mathcal{X}$, the oracle has $(100 - n)\%$ chance to return the correct preference (i.e., $r^*(y_w|x) > r^*(y_l|x)$), but has $n\%$ chance to return the incorrect/flipped preferences. We can control the noise rate of this oracle by adjusting the value of $n$.

- **Stochastic Noise Oracle**. For a prompt $x$ and two responses $y_w, y_l$, Stochastic Noise Oracle prefers $y_w$ over $y_l$ with the probability $\sigma[(r^*(y_w) - r^*(y_l))/\gamma]$, where $\sigma$ is the sigmoid function, and $\gamma \in \mathbb{R}^+$ is the temperature hyperparameter. We can control the noise rate by tuning the $\gamma$ value: The higher the $\gamma$ value, the more unpredictable the oracle is, and hence more noisy the preferences will be.

- **Gaussian Noise Oracle**. Stochastic Noise Oracle requires access to the gold reward model, which is infeasible in practice. Gaussian Noise Oracle, instead, only requires the access to an approximated reward model $r'$, such that $r'(y|x) = r^*(y|x) + \epsilon$, where $\epsilon$ is the *noise term* drawn from a Gaussian distribution $\mathcal{N}(\mu, \delta^2)$. The preference directions are then derived from the approximated reward $r'$. With $r'$, the probability of Gaussian Noise Oracle prefers $y_w$ over $y_l$ is:

$$\begin{aligned}p(y_w > y_l|x) &= \mathbb{1}[r'(y_w|x) - r'(y_l)] \\ &= \mathbb{1}[(r^*(y_w|x) + \epsilon_w) - (r^*(y_l|x) + \epsilon_l)] \\ &= \mathbb{1}[(r^*(y_w|x) - r^*(y_l|x)) + (\epsilon_w - \epsilon_l)].\end{aligned}$$

Since both $\epsilon_w$ and $\epsilon_l$ are drawn from the same Gaussian distribution $\mathcal{N}(\mu, \delta^2)$, $\epsilon_w - \epsilon_l$ is a random variable drawn from $\mathcal{N}(0, 2\delta^2)$. Hence, the noise rate of Gaussian Noise Oracle can be adjusted by tuning the value of $\delta$.

We believe the three strategies cover some widely observed noise types in preferences. For example, Stochastic Noise Oracle is also known as *Boltzmann rational* (Ziebart et al., 2008; Jeon et al., 2020; Gao et al., 2020) and widely used for simulating noise in human-provided preferences caused by *aleatoric uncertainty* (Hüllermeier & Waegeman, 2021). Gaussian Noise Oracle simulates the noise caused by the *epistemic uncertainty*, i.e., the RMs fail to accurately approximate the human's preferences. Random Noise Oracle simulates the random mistakes observed in both human-provided (Lindner & El-Assady, 2022) and heuristic-based preferences (Chen et al., 2024). We note that in reality, multiple types of noise can co-exist in the preference pairs; however, to ease analyses, in this paper, we assume there exist at most one type of noise in the preferences. We leave mixed-type preference noise for future work.

## 5 Experimental Setup

**Tasks.** We consider preference-based GLM alignment on two tasks: Reddit TL;DR (Stiennon et al., 2020) and Anthropic-Helpful (Bai et al., 2022a). Reddit TL;DR has two subsets, a SFT set and a preference set. In its preference set, each data entry consists of a document $x$ and two candidate summaries $y_w, y_l$ for $x$. It has 93k/53k/33k data entries in train/validation/test splits. Anthropic-Helpful is a subset of the AnthropicHH dataset, in which each data entry contains the dialogue history between a human and an AI assistant ($x$), and two candidate responses ($y_w, y_l$). It has 161k/9k data entries in train/test splits, and we separate out 1k randomly-sampled entries from the train set as the validation set.

**Generative Language Model.** For each task, we fine-tune a T5-Large (770M parameters) model to obtain the initial GLM $\pi_{\text{sft}}$. For TL;DR, we use the SFT subset in Reddit TL;DR as the SFT training data, which has 117k/6k/6k examples in train/validation/test splits. For Anthropic-Helpful, we use all the preferred responses in the dataset as the SFT training data. All the hyperparameters used in SFT are the same as in (Liu et al., 2024).

**Gold Reward Model $r^*$.** For each task (TL;DR and Anthropic-Helpful), we train a T5-XXL (11B parameters) (Raffel et al., 2020) model with the respective preference pairs to build the gold reward model $r^*$. In line with (Zhao et al., 2023), we format the input to the model with the prompt: [CONTEXT] $\{x\}$ [RESPONSE] $\{y\}$, and use the logit of the token 1 as a point-wise score for the reply.

**Noisy Preference Oracles.** Based on the gold reward model $r^*$ described above, we build the noisy preference oracles as described in §4. The noise rate of different noisy preference oracles can be controlled by tuning their respective hyperparameters: $n$ (noise rate) for Random Noise Oracle, $\gamma$ (temperature) for Stochastic Oracle, and $\delta$ (standard deviation) for Gaussian Noise Oracle. To decide the exact values of the hyperparameters for a target noise rate (e.g., 20%), we randomly sample 1k examples from each train set, and increase the hyperparameters with a small step (0.01) until the target noise rate is reached. The final hyperparameter values are presented in Table 2.

|  |  | 5% | 10% | 15% | 20% | 25% | 30% | 35% | 40% | 45% | 50% |
|---|---|---|---|---|---|---|---|---|---|---|---|
| TL;DR | $\gamma$ | 0.2 | 0.4 | 0.65 | 0.9 | 1.25 | 1.75 | 2.50 | 3.90 | 8.0 | 100 |
|  | $\delta$ | 0.34 | 0.70 | 1.09 | 1.55 | 2.15 | 2.90 | 4.05 | 6.50 | 13.0 | 100 |
| Anthropic | $\gamma$ | 0.11 | 0.22 | 0.36 | 0.53 | 0.75 | 1.06 | 1.54 | 2.45 | 4.95 | 100 |
|  | $\delta$ | 0.18 | 0.40 | 0.62 | 0.89 | 1.25 | 1.75 | 2.49 | 4.00 | 8.75 | 100 |

Table 2: Hyperparameters for the Stochastic Noise ($\gamma$) and Gaussian Noise Oracle ($\delta$) at each target noise rate (column). The hyperparameter for Random Noise Oracle ($n$; see §4) is omitted, as it equals the corresponding target noise rate.

**Generation Pairs.** In line with Liu et al. (2024), we use the trained GLM $\pi_{\text{sft}}$ to sample responses for each prompt, and pair up the sampled responses to build the Generation Pairs

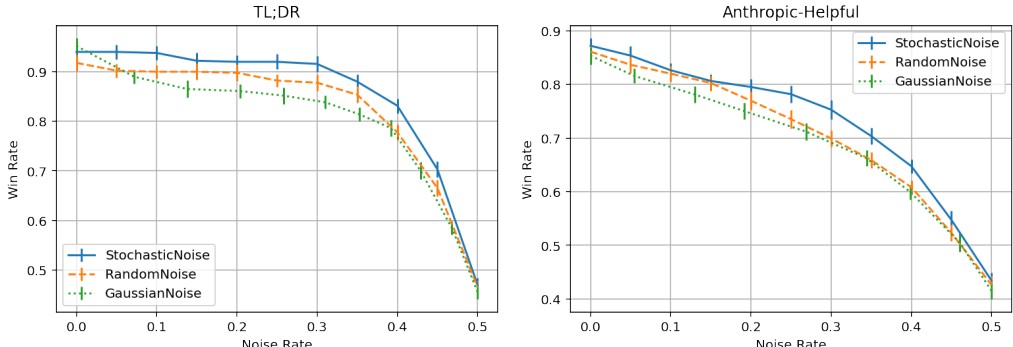

Figure 2: Influence of the noise rate (x-axis) on the alignment performance (in terms of the win rate; y-axis). Error bars: 95% confidence intervals, computed with double-tailed t-test on 5-10 repeated experiments with different random seeds.

dataset (see Fig. 1). For each prompt $x$, we sample eight generations from $\pi_{\text{sft}}(\cdot|x)$ with temperature 0.7, and randomly group them into four pairs. The pairs are then presented to the noisy preference oracles to build the Preference Pairs dataset (see Fig. 1).

**Other Hyperparameters.** We choose the hyperparameters by following the choices made in (Rafailov et al., 2023; Liu et al., 2024): in training the gold reward model and GLM $\pi_{\text{sft}}$, we use batch size 32 and learning rate 1e-5 with Adafactor optimizer (Shazeer & Stern, 2018); in the alignment training stage (Eq. (2)), we use $\beta = 0.5$, and dropout rate 0.1. Later in §7, we will explore different values for the regularization weights ($\beta$ and dropout rate) to study their effectiveness in mitigating the negative effect of noisy preferences.

## 6   Impact of Noise Rates on Alignment Performance

Fig. 2 presents how the alignment performance changes with the growth of the noise rates. We make the following observations.

- **Alignment performance drops with more noise in preferences.** This applies to all types of noise and both tasks we have considered. Also, we note that the alignment performance drops more quickly with the increase of the noise rates: When the noise rates are below 0.3, an increase of 10 percentage points (pp) in noise rate yields less than 10pp drop in the alignment performance; however, when the noise rates are higher than 0.4, 10pp increase in noise can yield 20-30 pp loss in performance.

- **Different types of noise cause similar harm.** At the same noise rate, the alignment performance of the three different noise types do not have significant differences, suggesting that it is the noise rate rather than the noise type that decides the alignment performance.

- **Alignment is beneficial even with highly noisy preferences.** In both tasks, the win rate is above 0.5 even with noise rate at 0.45. This observation reaffirms the effectiveness of alignment training (Touvron et al., 2023), and explains why even highly noisy preferences are used in alignment training in practice (see Table 1). But it is also worth noting that when the preferences are completely random (i.e., noise rate at 50%), the win rate drops below 0.5, suggesting that alignment is detrimental with random preferences.

## 7   Mitigate the Negative Impact of Preference Noise

In this section, we explore two popular strategies to mitigate the negative impact of preference noise: *regularization* in §7.1 and *data filtering* in §7.2.

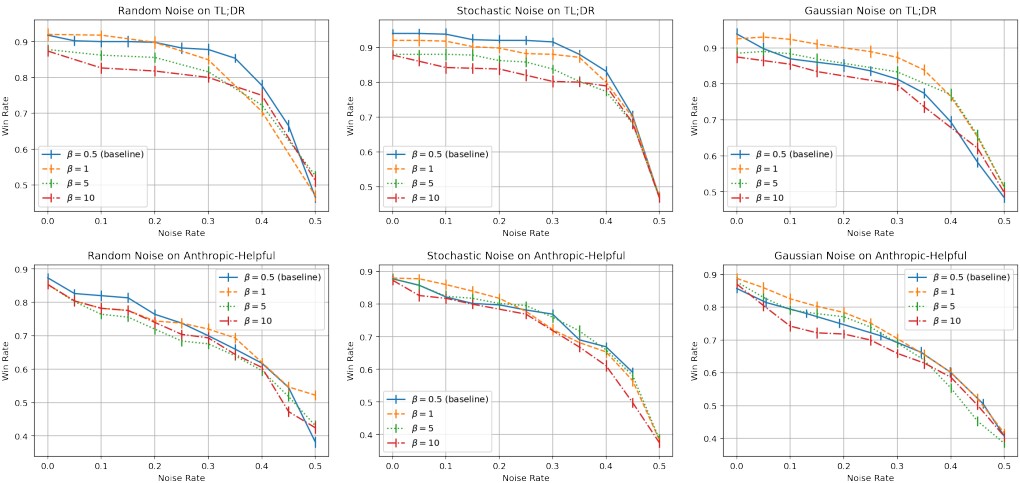

Figure 3: Alignment performance with different KL regularization weights $\beta$.

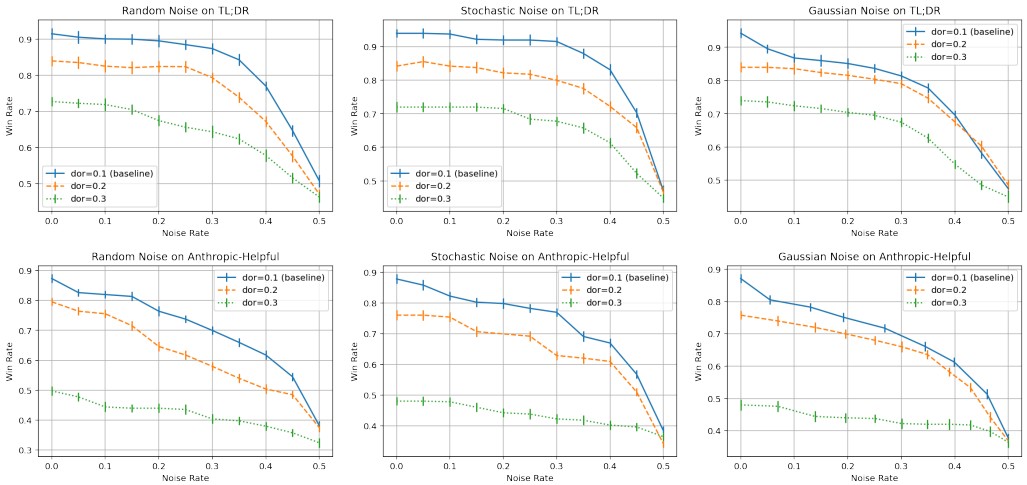

Figure 4: Alignment performance with different dropout rates (dor).

## 7.1 Regularization

We consider two methods to strengthen regularization: increasing the weight of the KL divergence loss ($\beta$ in Eq. (2)), and increasing the dropout rate.

Fig. 3 presents the alignment performance with different strengths of KL regularization. In general, we find that stronger KL regularization fails to mitigate the negative impact of preference noise. In some cases (e.g., Random and Stochastic Noise in TL;DR), higher strength of KL regularization even hurts the performance. Our finding reaffirms the limitations of KL-based regularization in DPO/SLiC (Azar et al., 2023).

Fig. 4 presents the alignment performance with different dropout rates. We find that higher dropout rates significantly harm the alignment performance. To summarize, our findings suggest that high strength of regularization, in general, fails to mitigate the negative impact of preference noise, and in certain cases can even hurt the alignment performance.

## 7.2 Data Filtering

Another approach to fight against noise is to filter out noisy data in the Preference Pairs dataset (see Fig. 1) and only use the remaining data to perform alignment. We use the popular *confidence-based* data filtering method (Cheng et al., 2008). For a prompt $x$ and two candidate responses $y_w, y_l$, the *confidence of $y_w$ preferred over $y_l$*, denoted $c(y_w > y_l|x)$, is a real value between 0 and 1. Ideally, the confidence function $c$ should be *well-calibrated* (Silva Filho et al., 2023), i.e., $c(y_w > y_l|x)$ is identical to the true probability of $y_w$ preferred over $y_l$. In confidence-based data filtering, only pairs with the confidence level larger than a pre-defined threshold $t$ are used in training; hence, if threshold $t = 0.5$, no filtering is performed; $t = 1$ means all data will be removed. We experiment with $t = 0.5, 0.6, \cdots, 0.9$ to investigate the effect of different levels of data filtering.

Based on the user studies made by Gao et al. (2020), we use the Bradley & Terry (1952) model to estimate the confidence value: $c(y_w > y_l|x) = \sigma[r^*(y_w|x) - r^*(y_l|x)]$, where $\sigma$ is the sigmoid function. In practice, there are multiple methods for estimating the confidence values, e.g., *conformal predictors* (Shafer & Vovk, 2008; Einbinder et al., 2022), *ensemble* methods (Liang et al., 2022), and *bias estimation* methods (Chen et al., 2023).

Note that with higher filtering threshold $t$, the *quality* of the remaining data is improved at the cost of *quantity* loss. We deliberately do not back-fill the filtered data, because in practice it can be prohibitively expensive to collect more preference pairs. This setup also allows us to study the trade-off between data quality and data quantity in preference-based alignment. Fig. 6 in Appendix A shows how the size of the remaining data shrinks with higher confidence thresholds. We find that in both datasets, the data size drops quite quickly with the growth of $t$ values: Almost 20% data are filtered as $t$ increases by 0.1.

Fig. 5 presents the alignment performance with different strengths of data filtering. We make the following observations.

- **Data filtering does not help to fight against Random Noise.** This is because Random Noise Oracle flips pairs *completely at random* (Frénay & Verleysen, 2014), i.e., the flipping chance of each pair is uniformly at random, not affected by any other factors (e.g., the prompt $x$ or the responses $y_w, y_l$). As a result, our confidence-based filtering cannot reduce the number of the noisy pairs in the filtered data, and hence fails to improve the performance.
- **Data filtering is effective to mitigate the harm from Stochastic and Gaussian Noise.** We find that with certain threshold (e.g., at 0.8), data filtering shows consistent and significant improvement across all noise rates, noise types, and tasks. Considering that over 50% preference pairs have been removed with data filtering at confidence threshold 0.8, this result suggests that data quality has a significant impact on the alignment performance.
- **Over-aggressive data filtering hurts the performance.** With very high confidence thresholds (e.g., 0.99), the alignment performance is compromised across all noise rates, noise types, and tasks, due to the severe loss of data size. Hence, it is important to properly trade off between data quality and data quantity, in order to yield the optimal alignment performance.

To better understand how data filtering improves the data quality, we investigate the noise rate before and after data filtering, with different noise types. Fig. 7 in Appendix A shows the noise rates in data filtered with different confidence thresholds. We find that when the noise is from Random Noise Oracle, the noise rate stays the same with all data filtering thresholds; this explains why data filtering does not help improve its performance. When the noise is from Stochastic or Gaussian Oracles, data filtering can effectively reduce noise rate, explaining the performance boost observed in Fig. 5.

## 8 Limitations & Future Work

**Generalizability.** We apply a popular alignment algorithm (DPO) to T5-based language models in our experiments. We believe our observations presented in §6 and §7 can be

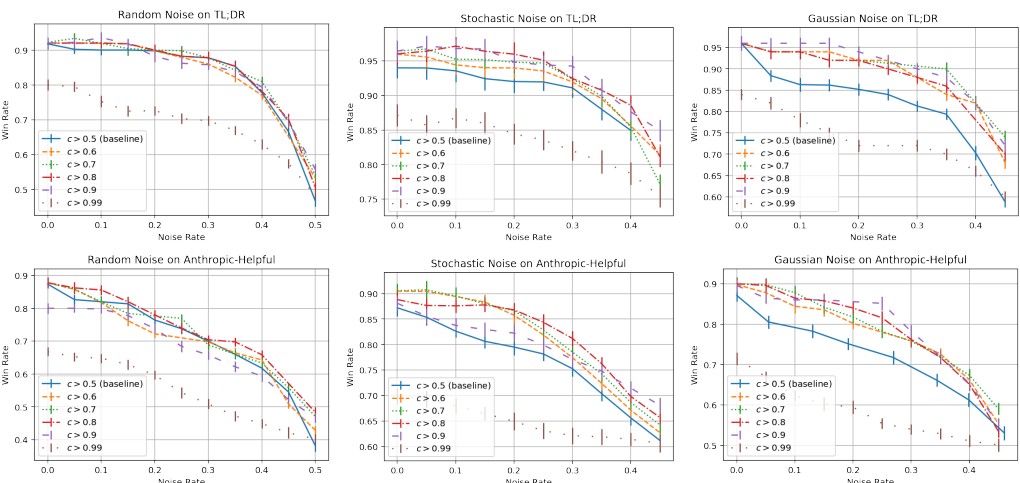

Figure 5: Alignment performance with confidence-based data filtering.

generalized to other alignment methods (e.g., PPO, SLiC and IPO, because they essentially optimize the same objective function; see §2), other Transformer (Vaswani et al., 2017) based language models, and other datasets (e.g., UltraFeedback (Cui et al., 2023) and AlpacaEval (Dubois et al., 2024)), but this is yet to be empirically verified. Considering the large number of possible combinations between alignment techniques and language models, a systematic study is beyond the scope of this paper, and we leave it for future work.

**Mixture of Noise.** We consider three strategies to add noise to the preferences, but we only allow one strategy to be used at one time (see §4). In practice, there may exist other types of noise, and different types of noise can co-exist in preferences. Our framework allows for creating new noise types by mixing primitive noise types, but it remains unclear whether the current observations can be applied to the new noise types or not. We hope our work can facilitate and encourage more work on this direction.

**Data Filtering.** Our current experiments demonstrate that data filtering can improve performance on the in-domain test set (see §7.2). However, it remains unclear whether this performance gain extends to out-of-domain test sets. A thorough understanding of data filtering's impact on cross-domain generalization requires meticulous experimental design. This includes careful consideration of factors such as the selection of training and testing tasks, and the degree of difference between them. We call for further research in this area. Additionally, the training set used and filtered in the current work is synthetic (see Section §5). Investigating the effectiveness of data filtering on human-generated datasets would be a valuable next step.

## 9  Conclusion

Pairwise preferences are widely used for aligning Generative Language Models (GLMs) with human values, but it remains unclear how the *noise in preferences* affect the alignment performance, and how to mitigate their negative impact. To study these problems, we propose a framework in which the types and rates of noise can be simulated and controlled, and we perform systematic experiments on two generation tasks (summary and dialogue generation) with this framework. Our findings suggest that alignment performance can be highly sensitive to the increase of noise rates, and appropriate data filtering is the most effective method to mitigate the negative impact of noisy preferences. Our work builds the first quantitative relation between noise rates and alignment performance across different noise types. We hope our work can help the community better understand and mitigate the impact of preference noise in GLM alignment.

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

# A   Appendix

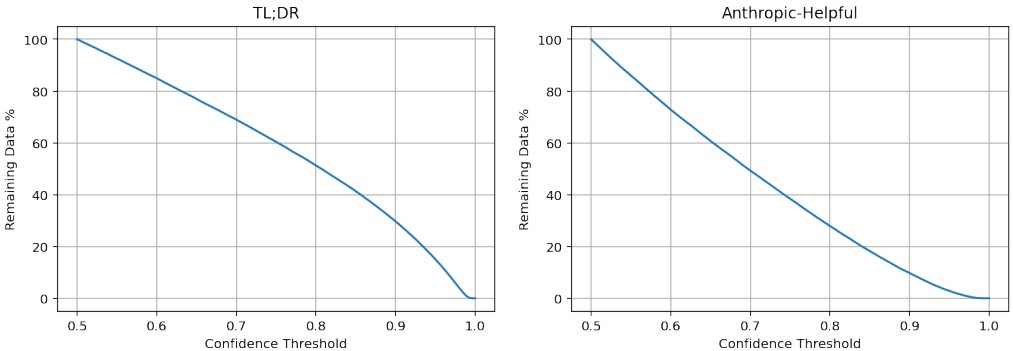

Figure 6: With higher data-selection threshold, fewer data will be remained for training.

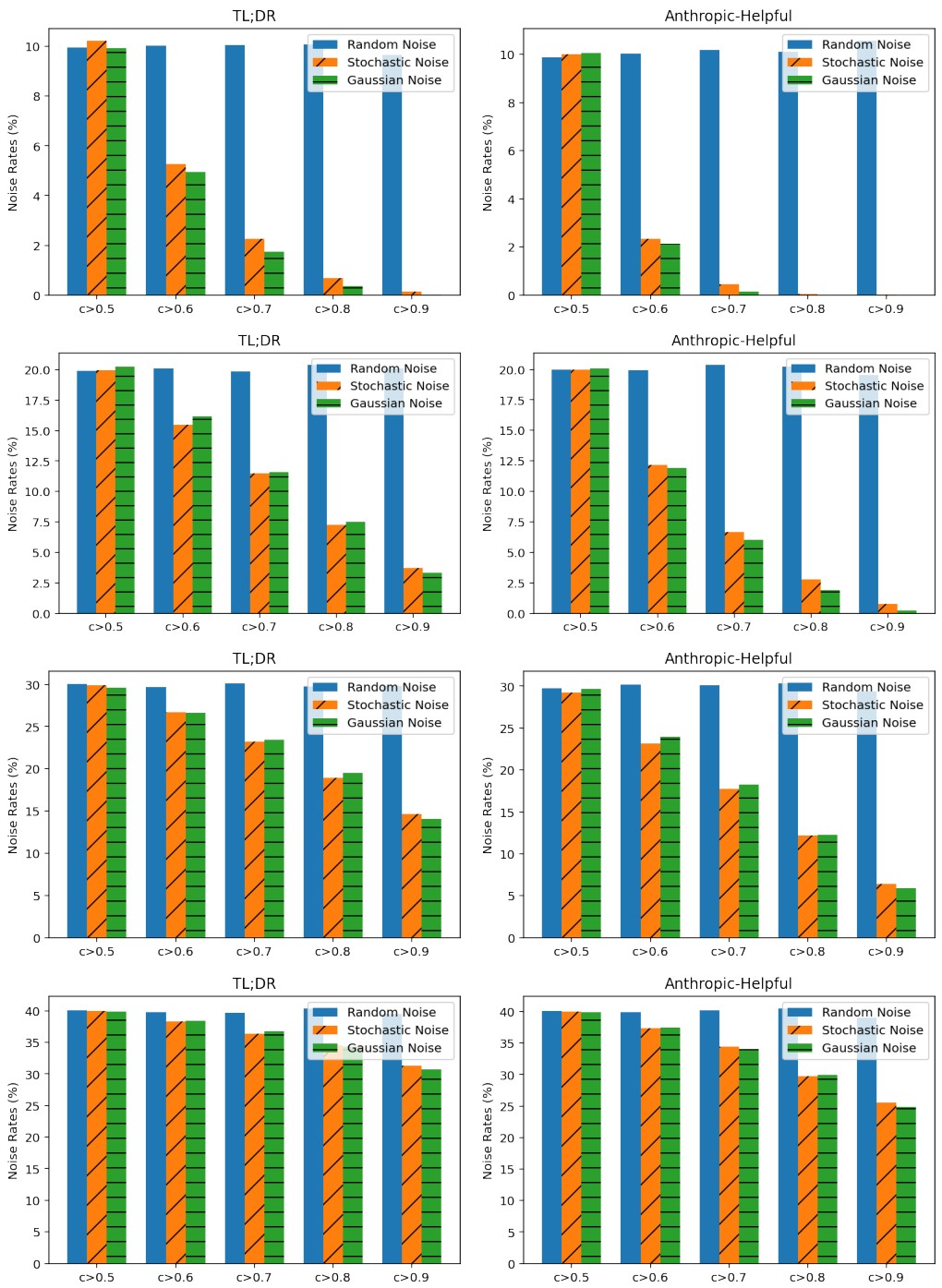

Figure 7: Noise rates of the filtered data, with different data filtering threshold. Note that when the threshold is 0.5, no data is filtered. Here the original/unfiltered data has 10% (first row), 20% (second row), 30% (third row), and 40% (fourth row) noise.

