# OpenReview forum: "Impact of Preference Noise on the Alignment Performance of Generative Language Models"
_colmweb.org/COLM/2024/Conference — COLM_

### Official Review · Reviewer_3A7y · 2024-05-07

**Rating:** 7
**Confidence:** 4
**Ethics Flag:** 1

**Summary:**

The paper studies a problem of noise in the preference alignment data for language models and its effect on the LLMs performance on 2 tasks, summarization and dialogue response generation. The authors use a controlled environment where they control the amount and the type of noise injected in the preference data and study the corresponding models' performance in terms of win rate of the aligned model vs the SFT model. The authors work with 3 types of noise: Random noise oracle (uniformly distributed noise across the dataset with a given ratio), Stochastic noise oracle (proportional to the reward difference between the preferred and dispreferred example) and Gaussian noise oracle (modelling the approximated rewards as stochastic gaussian variables).
Having found that noise in preference data does impact the alignment performance (but aligning even on the highly noisy data is beneficial for the model), the authors also conduct a study of performance degradation mitigation strategies in the settings of noisy preferences. Out of the two techniques they try, KL-based regularization and confidence-based data filtering, the latter results in superior robustness while not weakening the learning performance.

**Reasons To Accept:**

A comprehensive focused study of the effect of preference noise on the LM performance is conducted
An interesting and useful result is presented on mitigating the harm of preference noise: while regularization does not always help and decreases the alignment efficiency, data selection based on confidence addresses most of the noise types studied

**Reasons To Reject:**

The paper has contributions on the analysis side (the experimental studies themselves), modeling side (model training technique with confidence data selection), and resource side (noise datasets/framework). Out of the 3, I see the analysis contribution the most substantial - if analysis papers are welcomed at the conference, that works as a reason to accept, if not - this is a reason to reject.

Another thing that would add much substance to this work and is missing from the paper in my understanding is trying the confidence-based data filtering on a real unaugmented dataset with a known presence of preference noise and evaluating the resulting performance difference with a human study.

---

> ### Author Rebuttal · Authors · 2024-05-31
>
> Thanks very much for your reviews and feedback.
>
> We agree that applying confidence-based data filtering on real unaugmented data is a crucial next step in understanding the practical applicability and effectiveness of our proposed method. We believe it would involve:
> * **Confidence Estimation**: Developing robust methods to accurately gauge the confidence level of each preference pair in a real-world dataset. Now, in our simulated experiments, we estimate the confidence by assuming that we have access to the gold reward, but in practice, the confidence estimation involves leveraging model uncertainty, human annotations, or a combination of both.
> * **Threshold Selection**: Determining the optimal confidence thresholds for filtering data, so as to trade off between data quality and quantity.
> * **Human Evaluation**: Conducting comprehensive evaluations with human experts to assess the impact of confidence-based filtering on the quality of model outputs.
>
> Given the scope and complexity of these steps, we believe that it warrants a dedicated investigation, potentially as a separate research paper. In our revised manuscript, we will explicitly acknowledge this as a limitation and highlight the need for further research in this direction.

---

> > ### Comment · Reviewer_3A7y · 2024-06-06
> >
> > Thanks, the authors' response makes sense to me. I keep my judgement of this paper being a good analysis paper that would be beneficial to have at the conference.

---

### Official Review · Reviewer_9Tmq · 2024-05-09

**Rating:** 7
**Confidence:** 2
**Ethics Flag:** 1

**Summary:**

The paper investigates the impact of preference noise on the alignment of GLMs and proposes confidence-based data filtering to mitigate this noise. In controlled experiments, three types of artificial noise are inserted on preferences in the tasks of summarisation and dialogue generation. The authors find that 1) even high noise rates an still be beneficial for alignment and 2) that, contrary to regularisation, confidence-based data filtering improves performance.
The paper presents a small but focused contribution, with a firmly-states research question. It doesn't only investigate noise, but proposes mitigation solutions. Even though the scope is limited to two tasks, one model, and one noise type at a time, it can provide initial insights into the effect of preference noise and its mitigation.

**Questions To Authors:**

- Please change the line format in the graphs so that the different factor can be discerned by the colour blind and when printing in grey scale.

**Reasons To Accept:**

- The paper is clearly-written, with a small but novel, focused contribution
- Controlled experiments with different noise types sufficiently answer the research question

**Reasons To Reject:**

- None, since the limitations are acknowledged in the paper.

---

> ### Author Rebuttal · Authors · 2024-05-31
>
> Thank you very much for the review and feedback. We appreciate your recognition of our contributions and the clear presentation of our work. We will address the formatting issue with the figures to ensure they are color-blind-friendly and accessible in grayscale. We will do this for all figures in the final version if the paper is accepted.

---

> > ### Comment · Reviewer_9Tmq · 2024-06-04
> > **Authors responded to my concerns**
> >
> > The authors have responded to my concerns sufficiently. All in all, this is a suitable paper for the conference, which presents a small focused contribution.

---

### Official Review · Reviewer_2iR8 · 2024-05-10

**Rating:** 6
**Confidence:** 4
**Ethics Flag:** 1

**Summary:**

This paper investigates the impact of noise in preference data on the alignment performance of Generative Language Models (GLMs) and proposes methods to mitigate this impact. The authors provide a systematic study across two tasks—summarization and dialogue generation—highlighting how even slight increases in noise can significantly degrade performance. They introduce a novel framework for injecting and controlling noise types and levels in training data, leading to a detailed empirical analysis.

**Questions To Authors:**

(1) The paper handles single types of noise separately. How might the proposed methods perform under conditions of mixed noise types, which are more likely in real-world settings?

(2) While data filtering improves alignment performance, could it also potentially lead to overfitting or reduced model robustness due to reduced training data variability?

(3) Could the authors elaborate on why increased regularization was not effective? Are there other regularization techniques that could potentially yield better results in combating preference noise?

**Reasons To Accept:**

（1）The paper is well-structured, and the topic is highly relevant given the increasing reliance on GLMs for various applications.

（2）The paper introduces a novel framework for simulating preference noise and rigorously evaluates the impact using well-defined metrics and experimental setups.

（3）The results clearly demonstrate the sensitivity of GLMs to preference noise and offer practical solutions for mitigating these effects, providing valuable insights.

**Reasons To Reject:**

(1) The experiments are limited to two specific tasks and one type of GLM (T5). Additional experiments with other models and tasks could help generalize the findings more broadly.

(2) While the paper presents data filtering as an effective mitigation strategy, the exploration of regularization methods could be expanded, and other potential strategies could be considered.

---

> ### Author Rebuttal · Authors · 2024-05-31
>
> Thanks very much for your reviews and feedback.
>
> * **Why a small model on two specific tasks**: Please refer to our responses to Reviewer zDfF.
> * **Mixed-Type Noise**: We completely agree that exploring mixed-type noise is a crucial next step, as we've emphasized in Section 8. The primary goal of this paper was to establish a robust framework for evaluating the impact of noise on alignment performance. To streamline our initial analysis, we focused on single noise types. A comprehensive study of mixed-type noise would ideally involve a thorough analysis of the types of noise in real-world preference datasets and their proportions, and it is beyond the scope of this work. We will further highlight the significance of this research direction in the revised manuscript.
> * **Potential Harms of Data Filtering**: In our current experiments, the training and testing data are drawn from similar domains, and our results demonstrate that appropriate data filtering does not negatively impact in-domain generalization. However, to thoroughly understand the effects of data filtering on cross-domain generalization would require meticulous experiment design, considering factors such as the selection of training and testing tasks, and the degree of difference between them. While beyond the scope of this paper, we will acknowledge this as a limitation and advocate for future work in this area.
> * **Why Regularization Doesn't Work**: We hypothesize that the two regularization methods we employed (KL divergence and Dropout) indiscriminately remove signals from both "useful" and "noisy" preferences. Consequently, while they mitigate the harm caused by noisy preferences, they also diminish the effectiveness of clean preferences. Regularization in deep learning is a complex field, and we only utilized the most common techniques in this paper. We will encourage the readers to try other regularization methods (e.g., the ones discussed in Section III-B in "Learning from Noisy Labels with Deep Neural Networks: A Survey", Song et al. 2022). Also, given the importance of Language Models and the prevalence of noisy data in RLHF, we will explicitly call for the development of more tailored regularization methods specifically designed for RLHF in future research.

---

> > ### Comment · Reviewer_2iR8 · 2024-06-07
> >
> > Thank you for your response. I will keep my original ratings.

---

### Official Review · Reviewer_zDfF · 2024-05-11

**Rating:** 7
**Confidence:** 4
**Ethics Flag:** 1

**Summary:**

Authors present an empirical investigation about (1) the influence of noise in preference data on the model performance after alignment fine-tuning, and (2) what strategies could alleviate the degradation. First they describe three ways of injecting noise into preference data produced by their gold reward model such that those ways align with real world noise we may find in datasets we use. Second, they present an analysis of performance degradation as the amount of noise go up across multiple tasks and noise schedules. Finally, they investigate training regularization as well as data filtering and conclude that some methods might alleviate degradation but not completely. However, they highlight that even noisy preferences may still improve model's alignment compared to the reference model.

**Questions To Authors:**

Following on rejection reasons:

- why did you choose T5 770M model as the main GLM model ? Especially given other way more up-to-date LLMs available.

- why did you choose reddit TLDR instead of some more up to date tasks / benchmarks e.g. open assistant 2? Especially since alignment is tightly connected to instruction following tasks rather than summarization.

**Reasons To Accept:**

* this is a valuable analysis that is helpful for LLM community working on alignment. Provided filtering technique might be adopted and extended in future work.

* paper has a clear presentation and is easy to read and understand

**Reasons To Reject:**

* GLM model authors consider is pretty small (<1B) w.r.t. the parameters count. It feels that authors chose T5 family of models for some reason for both RM and GLM. Experimental results could have larger scope if e.g a llama3 8B or llama2 7B (if larger models are infeasible to train for authors) have been considered.

* general instruction following benchmarks such as alpaca eval is not considered in the experiments while it would be interesting to see the trends on winrates when numbers would correspond to some leaderboard.

---

> ### Author Rebuttal · Authors · 2024-05-31
>
> Thanks very much for your reviews and feedback.
>
> * **Why a small model: We acknowledge the limitations of using a smaller model (T5 770M).** This choice was made in line with recent work (Liu et al. 2024, “​​Statistical rejection sampling improves preference optimization”), and due to computational constraints. To conduct our statistical analysis rigorously, numerous experiments were required, making this the only model feasible within our resources. We recognize the value of exploring larger models, and will further highlight the significance of this research direction in the revised manuscript..
> * **Why TLDR and Anthropic-Helpful**: These datasets were chosen for their prevalence in preference alignment studies. While TLDR focuses on summarization, the Anthropic-Helpful dataset includes a substantial number of instruction-following examples (c.f., Bai et al. 2022, “Training a Helpful and Harmless Assistant with Reinforcement Learning from Human Feedback”), aligning with the instruction-following aspect of alignment. We agree that running experiments and evaluations on OpenAssistant2 and AlpacaEval would offer valuable insights. We will acknowledge this limitation of the current work, and encourage future research in this direction.

---

> > ### Comment · Reviewer_zDfF · 2024-06-05
> >
> > thanks for your reply!
> >
> > authors have addressed my comments, although i still think they could have trained at least 1.5B or 7B/8B llama2/3. There is a fair amount of frameworks and code that allows to train such models within 8 gpus or even less one uses parameter-efficient tuning.
> >
> > such training would help to see how this findings change when you scale the model up or when you use instruct based mode as the initialization etc. that is indeed a solid future work direction though

---

### Decision · Program_Chairs · 2024-07-10

**Decision:**

Accept

**Comment:**

This paper investigates the impact of noise in preference data on the performance of LLMs after alignment fine-tuning. It explores three noise injection methods and analyzes their effects on summarization and dialogue generation tasks. The study finds that noise degrades performance but aligning with noisy preferences can still improve over a reference model. It also evaluates mitigation strategies, concluding that confidence-based data filtering provides better robustness without compromising learning performance.

The reviewers appraise the paper being well-written and easy-to-follow, the method to inject noises being interesting and novel, and the comprehensive analysis being informative and instrument for the community.